# Role of *Sox3* in Estradiol-Induced Sex Reversal in *Pelodiscus sinensis*

**DOI:** 10.3390/ijms25010248

**Published:** 2023-12-23

**Authors:** Tong Zhou, Jizeng Cao, Guobin Chen, Yubin Wang, Guiwei Zou, Hongwei Liang

**Affiliations:** 1Yangtze River Fisheries Research Institute, Chinese Academy of Fisheries Science, Wuhan 430223, China; zhoutong@yfi.ac.cn (T.Z.); 15832942914@163.com (J.C.); cgb1251877642@126.com (G.C.); wybingo992@126.com (Y.W.); zougw@yfi.ac.cn (G.Z.); 2College of Fisheries and Life Science, Shanghai Ocean University, Shanghai 201306, China

**Keywords:** gonadal differentiation, *Pelodiscus sinensis*, *Sox3*, RNAi, sex reversal

## Abstract

The Chinese soft-shelled turtle *Pelodiscus sinensis*, an economically important species in China, exhibits significant sexual dimorphism. Males are more valuable than females owing to their wider calipash and faster growth. Estradiol (E2)-induced sex reversal is used to achieve all-male breeding of turtles; however, the mechanism of this sex reversal remains unclear. In this study, we characterized the *Sox3* gene, whose expression level was high in the gonads and brain and exhibited significant sexual dimorphism in the ovary. During embryonic development, *Sox3* was highly expressed at the initiation of ovarian differentiation. E2 and *Sox3*-RNAi treatment before sexual differentiation led to 1352, 908, 990, 1011, and 975 differentially expressed genes in five developmental stages, respectively, compared with only E2 treatment. The differentially expressed genes were clustered into 20 classes. The continuously downregulated and upregulated genes during gonadal differentiation were categorized into Class 0 (*n* = 271) and Class 19 (*n* = 606), respectively. KEGG enrichment analysis showed that *Sox3* significantly affected sexual differentiation via the Wnt, TGF-β, and TNF signaling pathways and mRNA surveillance pathway. The expression of genes involved in these signaling pathways, such as *Dkk4*, *Nog*, *Msi1*, and *Krt14*, changed significantly during gonadal differentiation. In conclusion, the deletion of *Sox3* may lead to significant upregulation of the mRNA surveillance pathway and TNF and Ras signaling pathways and downregulation of the Wnt and TGF-β signaling pathways, inhibiting E2-induced sex reversal. These findings suggest that *Sox3* may play a certain promoting effect during E2-induced sex reversal in *P. sinensis*.

## 1. Introduction

The Chinese soft-shelled turtle *Pelodiscus sinensis* is an economically important aquatic animal, widely distributed in East Asian countries such as China, Korea, and Thailand [1]. In 2023, the China Fishery Yearbook reported that the production of turtle meat in China reached 374,000 tons, with an output value of more than 20 billion yuan. Chinese soft-shelled turtle has become one of the favorite aquaculture species in China due to its excellent nutritional and medicinal values [2]. *P. sinensis* exhibits distinct sexual dimorphism; males are larger than females and have a wider calipash and are thus more popular in aquaculture and have a higher market value [3]. To reduce the production cost and feed waste and improve the revenue of aquaculture farms, it is of great significance to manipulate the sex of turtles and obtain all male offspring in the breeding process. Therefore, analysis of the molecular mechanism of sex reversal and all-male breeding of *P. sinensis* are important for the development of turtle fisheries [4].

In *P. sinensis*, sex is determined on the basis of a ZZ/ZW sex determination system [1], which is characterized by the presence of heterotypic chromosomes [5,6]. Sex is usually determined by a single gene or combination of genes on the sex chromosomes or autosomes [5]. Several genes such as SRY-box transcription factor 9 (*Sox9*) and Wnt family member 4 (*Wnt4*) are critical during sexual differentiation [7,8]. Liang et al. developed sex-specific markers to accurately identify the sex of *P. sinensis* using RAD-Seq. A previous study reported that when *P. sinensis* eggs were incubated at 30 °C, sexual differentiation began on day 16 (stage 13) and ended on day 22 (stage 20) [9]. Treatment with exogenous hormones such as estradiol (E2) and methyltestosterone (MT) during sexual differentiation induce sex reversal in *P. sinensis* [3]. It has been reported that exogenous estrogen treatment significantly affects sexual differentiation and embryonic development in turtles [10] and significantly upregulates the female sex-related genes Cytochrome P450 family 19 subfamily A member 1 (*Cyp19a1*) and R-spondin 1 (*Rspo1*) [11,12]. E2 induces sex reversal in male embryos (ZZ), leading to the development of pseudo-female individuals (∆ZZ) with a male genotype and female phenotype. The pseudo-female individuals (∆ZZ) can be crossed with mature male individuals (ZZ) to obtain all-male offspring, thus improving the profits of the turtle farming industry [12].

Sex determination is a complex process that involves many genes and hormones [13]. The mammalian male sex determination gene, sex-determining region of Y chromosome (*Sry*), contains a highly conserved high mobility group (HMG) box DNA-binding domain, and *Sry* gene deletion in male (XY) mice may cause them to develop into females [14,15]. In addition, *Sry* facilitates the development of primordial gonads into testes [16] and inhibits the female developmental pathway, thereby inhibiting ovarian development [17,18]. Evolutionary studies indicate that *Sry* is a hybrid of the first exon of the DiGeorge syndrome critical region gene 8 (*Dgcr8*) and SRY-box transcription factor 3 (*Sox3*), which belongs to the ancient *Sry*-related HMG box (*Sox*) gene family [19]. The Sox transcription factors share more than 50% amino acid sequence homology with the HMG box DNA-binding domain encoded by the *Sry* gene [20]. Unlike the *Sry* gene, which is found only in mammals, *Sox* genes are present in numerous species. According to the homology and protein structure of the HMG box DNA-binding domain, *Sox* genes are divided into the subgroups A–J [21].

The *Sox* family genes regulate many developmental processes, such as sexual differentiation, embryogenesis, nervous system development, and chondrogenesis, by activating or repressing target gene expression [22]. *Sox3*, a member of the subgroup *SoxB1*, is widely involved in the early regulation of neural stem cell differentiation [23,24] and plays an important role in embryonic development [25,26] and the determination of reproductive fate [17,27,28,29]. *Sry* inhibits the expression of *Sox3* in mouse primordial gonads, which promotes the expression of *Sox9*, leading to testis development [17]. Similarly, *Sox3* affects conversely developmental pathways like *Sry*. In the determination of reproductive fate in mammals such as mice, *Sox3* has been reported to inhibit the expression of *Sox9* in the ovaries, promote ovarian development, and even directly activate the transcription of the female sex-related gene *Cyp19* [27,28,30]. In fish, the role of *Sox3* in sex determination and gonadal differentiation varies with species. *Sox3* initiates male sexual differentiation in *Oryzias dancena* [28] and is involved in testicular differentiation and development and spermatogenesis in *Acanthopagrus schlegeli* [31] and *Clarias batrachus* [32,33]. In contrast, in other fish species such as *Oreochromis niloticus* [34] and *Epinephelus coioides* [29], *Sox3* is involved in ovarian development and oogenesis. In amphibian species such as *Rana rugosa* [27] and *Xenopus laevis* [30], *Sox3* inhibits *Sox9* in the ovaries, promotes ovarian development, and directly activates the transcription of *Cyp19*. These findings suggest that *Sox3* plays a vital role in gonadal differentiation in many species. However, the functional role of *Sox3* in reptiles such as *P. sinensis* remains unclear, mainly owing to the lack of systematic studies.

Transcriptomic analysis of turtle gonads has shown that *Sox3* is upregulated in estrogen-induced pseudo-female turtles (ΔZZ) and significantly inhibited in males (ZZ), suggesting that *Sox3* is an important female sex determination gene in *P. sinensis* [35]. In the present study, we investigated the function of *Sox3* in the sex reversal process using E2 and *Sox3*-RNAi treatment and performed transcriptomic analysis at different time points during sexual differentiation. The results indicated that *Sox3* plays a catalytic role in sex reversal and provided novel insights into the mechanism of sex reversal in reptiles, whose mode of sex determination is genotypic. These findings can lay a theoretical basis for all-male breeding in *P. sinensis*.

## 2. Results

### 2.1. Characterization of P. sinensis Sox3 Gene

RACE was used to clone the full-length sequence of *P. sinensis Sox3* from the ovarian tissue. The results showed that *Sox3* is 1922 bp in length, with an open reading frame of 933 bp, which encodes 310 amino acids (Figure 1A). Moreover, *Sox3* contained two conserved putative functional domains, an HMG-type DNA-binding domain and a SOXp motif (Appendix A). The *Sox3* of *P. sinensis* shared high amino acid sequence homology with *Sox3* of *Chelonia mydas* (XM_037908813.1) and *Chrysemys picta bellii* (XM_005294617.2; 92.63% and 91.72%, respectively). It shared the lowest amino acid sequence homology with its mammalian counterparts, that is, 57.5% with *Mus musculus Sox3* (NM_009237.2) and 56.95% with *Homo sapiens Sox3* (NM_005634.3). The phylogenetic tree showed that *P. sinensis* was closely related to other reptiles, followed by birds, fish, and amphibians (Appendix A). The functional structure of *Sox3* was similar to that of SRY-box transcription factor 2 (*Sox2*), and both belong to the *SoxB1* subfamily (Appendix A).

The male turtles were significantly larger than the female individuals, and both had been maintained for 2 years (Figure 1B). To investigate whether *Sox3* is involved in sexual differentiation and gonadal development in *P. sinensis*, qRT-PCR was used to analyze the expression patterns of *Sox3* in different tissues and embryonic development stages. *P. sinensis Sox3* was almost undetectable in the heart, liver, spleen, lung, kidney, intestine, and muscle but was highly expressed in the brain and gonads (Figure 1C). *Sox3* expression showed no sexual dimorphism in the brain but showed female biased expression in the gonadal tissue. The mRNA expression level of *Sox3* in the ovary was higher than that in the testis (*p* < 0.01). 

Furthermore, the mRNA expression levels of *Sox3* at different developmental stages were examined. *Sox3* expression showed a trend of an initial increase with a subsequent rapid decrease (Figure 1D). In females, the expression levels peaked at stage 11 before the initial stage of sexual differentiation (stage 12), but *Sox3* expression decreased rapidly or was even absent after stage 14. In males, *Sox3* was continuously expressed at low levels and showed almost no expression during gonadal development. *Sox3* expression showed sexual dimorphism from stages 9 to 13, that is, it was more abundant in female embryos. After stage 14, it showed low expression and no sex difference. These results indicate that *Sox3* has a sexually dimorphic expression pattern in *P. sinensis*, and it is expressed before sexual development initiation and in the early stage of gonadal differentiation.

### 2.2. Molecular Response of Sox3 to E2-Induced Male-to-Female Sex Reversal

To understand the role of *Sox3* in sex reversal of *P. sinensis*, 5 µL of 10 mg/mL E2 was injected into eggs at developmental stage 13 using a microinjector (Figure 2A). In the E2-induced group, the expression levels of *Sox9* at various developmental stages of male embryos were significantly lower than that in the untreated group (Figure 2B). However, the expression levels of Wnt4 were significantly higher than the expression levels in the control group during different developmental stages (Figure 2C). The expression of *Sox3* began to increase significantly at developmental stage 13 and reached the maximum at stage 18. Overall, the relative expression levels of Sox3 in the E2-induced group were significantly higher than those in the control group (Figure 2D). Moreover, the expression levels of *Sox3* in female and pseudo-female adult *P. sinensis* were significantly higher than those in male adult *P. sinensis* (Figure 2E).

### 2.3. Molecular Response of Male Embryos with Sox3 Knockdown

To further explore the molecular role of *Sox3* in the sex reversal process, we identified three target sequences on the *Sox3* gene that were used to synthesize three distinct lentiviral interference vectors (Figure 3A and Appendix A). The expression trend of *Sox3* in male embryos was targeted by injecting a *Sox3* lentiviral interference vector in combination with E2 into embryos at the beginning of sexual differentiation (Figure 3B). The lentiviral interference vector constructed in this experiment had a significant interference effect on the expression of *Sox3* in male embryos during the sex reversal process. The expression level of *Sox3* in male embryos after E2 and Lv-*Sox3*-765 induction was significantly lower than that in the control embryos (injected with E2 and Lv-NC; Figure 3C). 

We also analyzed the expression of *Sox9* and *Wnt4*, which are important in the male and female developmental pathways, respectively. In male embryos injected with E2 and Lv-*Sox3*-765, the mRNA expression levels of the testicular marker *Sox9* was significantly higher than those in control (CK) embryos at stages 14 to 18 during gonadal differentiation periods (Figure 3D).

### 2.4. Transcriptome Analysis at Various Gonadal Differentiation Stages during Sex Reversal of P. sinensis after Sox3 Knockdown

Embryonic samples at various gonadal differentiation stages were collected from groups that were induced with E2 and *Sox3*-RNAi (*Sox3* group: *Sox3*-1, *Sox3*-2, *Sox3*-3, *Sox3*-4, and *Sox3*-5) and induced with only E2 (CK group: CK-0, CK-1, CK-2, CK-3, CK-4, and CK-5; Figure 4A). The similarity among the three biological replicates was tested by principal component analysis; the results showed high similarity among the replicate samples (Figure 4B), indicating that the sequencing data can be further analyzed. The reads align region and the expression violin plot of all samples showed that the sequencing quality of all samples was good enough for further analysis (Appendix A). In the five *Sox3*-RNAi groups, 15,920, 15,911, 15,772, 15,888, and 15,994 genes were found to be expressed, respectively. In the five CK groups treated by E2, 15,932, 15,372, 15,685, 15,816, 15,925, and 15,877 were found to be expressed, respectively (Figure 4C). A comparison of the genes identified between the corresponding *Sox3*-RNAi group treated by E2+*Sox3*-RNAi and the CK group treated by E2 at each developmental stage identified 1352, 908, 990, 1011, and 975 differentially expressed genes, respectively (Figure 4D). The differentially expressed genes are composed of CK group specific genes, *Sox3*-RNAi group specific genes, and overlapping genes at each stage. The specially expressed genes in the CK group induced by E2 at different developmental stages were compared and we found no common gene existed in all five periods; this also occurred in the *Sox3*-RNAi group (Appendix A). The specially expressed genes that existed in at least four periods in the CK group or *Sox3*-RNAi group are listed in Appendix A.

### 2.5. Clustering and KEGG Enrichment Analysis after Sox3-RNAi Treatment

Trend analysis was performed on all genes in the CK and *Sox3* groups identified as expressed (Figure 5A,B). The results showed that the expressed genes in the CK groups were categorized into 20 distinct clusters. Of these, Cluster 0 (*n* = 3499) comprised genes that were continuously downregulated during gonadal differentiation; KEGG enrichment analysis showed that these genes were enriched in ubiquitin-mediated proteolysis, mRNA surveillance pathway, dopaminergic synapse, spliceosome, and cell growth and death. In contrast, Cluster 19 (*n* = 2604) comprised genes that were continuously upregulated during gonadal differentiation and were enriched in arachidonic acid metabolism, protein digestion and absorption, hematopoietic cell lineage, ECM–receptor interaction, and complement and coagulation cascades. Moreover, the genes that were upregulated in the early stage of gonadal differentiation and downregulated in the late stage were classified into Class 16 (*n* = 732) and Class 18 (*n* = 1387; Figure 5C). 

The expressed genes in the *Sox3* groups were also categorized into 20 distinct clusters. Cluster 0 (*n* = 2204) included genes that were continuously downregulated during gonadal differentiation; KEGG enrichment analysis showed that these genes were associated with Hippo signaling pathway, glutamatergic synapse, neuroactive ligand–receptor interaction, regulating pluripotency of stem cells, and axon guidance. In contrast, Cluster 19 (*n* = 2200) comprised genes that were continuously upregulated during gonadal differentiation and were enriched in cell adhesion molecules, hematopoietic cell lineage, protein digestion and absorption, ECM–receptor interaction, and complement and coagulation cascades. Class 16 (*n* = 527) included genes that were upregulated in the early stage of gonadal differentiation and downregulated in the late stage and were associated with the Wnt signaling pathway and cell cycle (Figure 5D).

### 2.6. Trend and KEGG Enrichment Analysis of Differentially Expressed Genes between the CK and Sox3 Groups

To further explore the role of *Sox3* during gonadal differentiation, trend analysis was performed on the differentially expressed genes at five time points between the corresponding CK group (treated with E2) and *Sox3* group (treated with E2+*Sox3*-RNAi) (Figure 6A). The results showed that the continuously downregulated genes were categorized into Class 0 (*n* = 271), and the continuously upregulated genes were categorized into Class 19 (*n* = 606). The top 15 downregulated and upregulated genes belonging to Class 0 or Class 19, respectively, are listed in Appendix A. The relative expression levels and related signaling pathways of the top 15 downregulated genes showed that *Sox3* affected the sexual differentiation of *P. sinensis* via pathways such as the Wnt signaling pathway, TGF-β signaling pathway, and NOD-like receptor signaling pathway. Of the DEGs between the CK group and *Sox3* group, the top 15 continuously downregulated genes, for example, *Dkk4*, *Nog*, *Mmp11*, *Ptchds*, *Cbx2*, and *Sppl3*, were associated with key signaling pathways such as the Wnt signaling pathway, TGF-β signaling pathway, and NOD-like receptor signaling pathway (Figure 6B). Some of the key upregulated genes were *Msi1*, *Krt14*, *Ccm1*, *Pla2g4c*, *Nox1*, and *Alox5*. Of these, *Pla2g4c*, *Nox1*, and *Alox5* were enriched in signaling pathways associated with ovarian development, which highlights the significant role of *Sox3* during gonadal differentiation (Figure 6C).

### 2.7. Validation of Key Genes Associated with Gonadal Differentiation Periods

*Sox3* deletion resulted in significant changes in the expression levels of many sex-related genes. qRT-PCR showed that the expression levels of some genes, such as *Dkk4*, *Nog*, *Mmp11*, *Ptchd3*, *Sppl3,* and *Cbx2*, continuously decreased throughout the gonadal differentiation periods (Figure 7A). In contrast, the expression levels of *Msi1*, *Krt14*, *Ranbp1*, *Npffr2*, *Mapk10*, and *Ccm1* continuously increased during gonadal differentiation (Figure 7B). The verification results suggest that *Sox3* deletion induced the activation of certain signaling pathways, such as the mRNA surveillance pathway, TNF signaling pathway, and Ras signaling pathway, and inhibited other signaling pathways, such as the Wnt signaling pathway, TGF-β signaling pathway, and NOD-like receptor signaling pathway, thus blocking the E2-induced sex reversal process in *P. sinensis* (Figure 7C).

## 3. Discussion

In this study, the *Sox3* gene of *P. sinensis* was cloned, and its expression characteristics in early embryonic development and expression changes in various stages of embryonic sexual differentiation after induction with E2 and *Sox3*-RNAi suggest that *Sox3* plays an important role in sex reversal in *P. sinensis*. Amino acid sequence and structure analysis showed that *Sox3* contains two conserved putative functional domains, an HMG-type DNA-binding domain and a SOXp motif. *Sry* has a highly conserved HMG box DNA-binding domain and is a key gene associated with sexual differentiation [20]. *Sox3* and *Sox2* have similar conserved domains. Both have been reported to activate the expression of geminin in naive *Xenopus* ectodermal explants, and *Sox3* activated the expression of *Sox2* but not vice versa [36]. *Sox3* and *Sox2* have equivalent functions in brain and testis development in mouse embryos [37]. Genes that contain highly conserved HMG box DNA-binding domains may play important roles in the process of sex differentiation in aquatic animals.

Estrogen promotes the development of the cortical tissue of primordial gonads and inhibits the development of the medullary tissue, which leads to female gonadal development [38]. In aquatic animals such as *Cynoglossus semilaevis* [39], *Acipenser sinensis* [40], and *C. mydas* [41], estrogen promotes ovarian differentiation and inhibits testis differentiation. A previous study also showed that E2 promotes ovarian development in *P. sinensis*, and high concentrations of E2 lead to male-to-female sex reversal [12]. In the present study, after E2 treatment, the expression levels of *Wnt4* and *Sox9* were significantly upregulated during gonadal differentiation. Moreover, the expression level of *Sox3* was significantly upregulated in male embryos but was very low in the gonads of adult male individuals. These findings suggest that *Sox3* plays a key role in the sex reversal process in *P. sinensis*.

Previously, most studies investigating the role of *Sox3* in sexual differentiation and embryonic development have been conducted in mammals, fishes, and amphibians, with few studies in reptiles. It has been reported that *Sry* inhibits the expression of *Sox3* in mouse primordial gonads, which promotes the expression of *Sox9*, leading to testis development [17]. Using *Sry* deletion, it was found that in chicken embryonic development, *Sox3* expression plays a role in both sexes and *Sox3* is expressed earlier than *Sox9* [42]. In the present study, after *Sox3* knockdown, *Sox9* expression was upregulated during sexual differentiation, which indicated *Sox3* and *Sox9* may play opposite roles in the process of sex differentiation of Chinese soft-shelled turtles.

It has been established that multiple signaling pathways work together to regulate the sex reversal process [43,44]. A previous study found that sexual differentiation of reptiles with genetic sex determination is closely associated with the RSPO1/WNT/β-catenin signaling pathway [45]. The sex reversal process also involves many components, such as the sex-determining region Y (*Sry*), SRY-box transcription factor 9 (*Sox9*), and fibroblast growth factor 9 (*Fgf9*) [46]. The CK group in Figure 4 was induced by E2 at stage 13 before sexual differentiation. The trend analysis showed the expression trend of genes in the sex reversal process. In the present study, KEGG enrichment analysis showed that key signaling pathways, such as ubiquitin-mediated proteolysis, cell growth and death, and the Wnt signaling pathway affect the process of sex reversal. These important signaling pathways may be upstream of the RSPO1/WNT/β-catenin signaling pathway and are activated or inhibited by estrogen receptors in different ways. However, the specific mechanism of action in *P. sinensis* is still unclear. Moreover, genes that were continuously downregulated or upregulated during gonadal differentiation (Class 0 and Class 19, respectively) in the *Sox3* group were enriched in the Hippo signaling pathway, ECM–receptor interaction, and complement and coagulation cascades, suggesting that these signaling pathways are closely associated with *Sox3* in the sex reversal process of *P. sinensis*.

*Sox3* is required for neurogenesis; it participates in the early regulation of neural progenitor cell proliferation and neural stem cell differentiation [47]. Furthermore, in most vertebrates, *Sox3* has been reported to be involved in sexual differentiation. On the one hand, it has been reported that *Sox3* initiates male sexual differentiation in *O. dancena* [28] and participates in testis development and spermatogenesis in *M. musculus* [48], *A. schlegeli* [31], and *A. schlegeli* [32]. On the other hand, *Sox3* is reported to be highly expressed in the ovary in species such as *R. rugosa* [27], *O. niloticus* [34], *Paralichthys olivaceus* [49], and *A. sinensis* [50]. Consistent with these findings, our results indicate that *Sox3* plays a role in female sex determination by affecting key genes in the Wnt signaling pathway.

In total, 1917 differentially expressed genes categorized into 20 classes were identified between the CK and *Sox3* groups at five developmental stages, which demonstrated the key role of *Sox3* in sex reversal (Figure 5). A total of 606 and 271 genes were observed to be continuously upregulated and continuously downregulated, respectively, during sexual differentiation. Their association with important signaling pathways, such as the Wnt signaling pathway, TGF-β signaling pathway, matrix metalloproteinase pathway, and estrogen signaling pathway indicates that significant changes occurred in these pathways. The expression levels of genes such as *Dkk4*, *Nog*, *Mmp11*, and *Gnao1* associated with these signaling pathways were significantly downregulated in the sex reversal process of *P. sinensis*. The deletion of *Sox3* also resulted in an increase in the expression of significant genes in male-related developmental pathways, such as *Msi1*, *Krt14*, and *Ranbp1*. These findings suggest that *Sox3* plays a promoting role in sex reversal, inhibiting the expression of male-related genes and promoting the expression of female-related genes.

## 4. Materials and Methods

### 4.1. Experimental Materials

Male and female turtles (2 years old) and fertilized eggs were obtained from Anhui Xijia Agricultural Development Co., Ltd. (Bengbu, Anhui, China). The fertilized eggs were incubated in egg incubators at 30 °C. Embryos at various developmental stages (stages 9–21) were sampled and stored in liquid nitrogen [51]. Adult turtles were anesthetized with 0.05% MS 222 (Sigma, St. Louis, MO, USA), and their tissues (heart, liver, spleen, lung, kidney, intestine, brain, muscle, and gonads) were collected and stored in liquid nitrogen.

### 4.2. Cloning of the P. sinensis Sox3 Gene

Total RNA was extracted from the ovary using TRIzol reagent, and the first strands of cDNA and RACE cDNA were synthesized using the HiScript^®^ III 1st Strand cDNA Synthesis Kit (+gDNA wiper) (Vazyme, Wuhan, China) and SMARTer^®^ RACE 5′/3′ Kit (Takara, Dalian, China), respectively. Specific primers for the conserved region of the *Sox3* gene were designed on the basis of the predicted sequence from transcriptome sequencing of *P. sinensis* (Appendix A). PCR amplification was performed using ovarian cDNA as a template; the PCR products were sequenced by Wuhan Tianyi Huayu Gene Technology Co., Ltd. (Wuhan, China). Next, specific primers for 5′ and 3′ RACE amplification were designed on the basis of the conserved sequence of the *Sox3* gene (Appendix A). The primer *Sox3*-3′-GSP1 and primer 3′ adapter from the kit were used for 3′ RACE, whereas the primer *Sox3*-5′-GSP and primer 5′ adapter from the kit were used for 5’ RACE and the cDNA with a tail was used as the template. The target band was purified and recovered, ligated into the pMD-18T vector (Takara), and transformed into DH5α competent cells (Biomed, Beijing, China) for culture. Positive clones were picked for sequencing.

The cloned fragments were spliced together using DNAMAN version 9.0 (LynnonBiosoft, Shanghai, China), and the open reading frames (ORFs) and amino acid sequences were predicted using NCBI ORF finder (https://www.ncbi.nlm.nih.gov/orffinder/, accessed on 18 January 2023). Then, SMART (http://smart.embl-heidelberg.de/, accessed on 20 May 2023) and Pfam (http://pfam.xfam.org/, accessed on 18 January 2023) were used to analyze the protein structure. The amino acid sequence homology was compared using DNAMAN version 9.0, and a phylogenetic tree based on *Sox3* was constructed by the neighbor-joining method using MEGA X.

### 4.3. Exogenous Estradiol Treatment

Exogenous E2 was injected into eggs at the developmental stage 13 (day 16). Stage 13 represents a crucial phase in the gonad development stage that precedes gonad differentiation when incubated at a constant temperature [52]. E2 powder was dissolved in anhydrous ethanol to achieve a final concentration of 10 mg/mL, and 5 µL was injected after corroding the egg shell with a small amount of hydrochloric acid. Previous studies showed that 5 µL of 10 mg/mL E2 was the most effective dose to induce sexual reversal in Chinese soft-shelled turtle [9]. The control group was injected with an equal volume of anhydrous ethanol. The E2-injected and control group eggs were further incubated at 30 °C. Embryos were collected at stages 13–19 after treatment and stored in paraformaldehyde or liquid nitrogen.

### 4.4. Sox3 Lentiviral Interference Vector-Injected Turtle Embryos

Three *Sox3* lentiviral interference vectors (*Sox3*-567, *Sox3*-765, and *Sox3*-986) and the blank vector LV-NC-shRNA were constructed (GenePharma, Suzhou, China). The primer sequences are listed in Appendix A. Using 293T cells, the titer of the lentiviral vectors was determined to be 1 × 10^9^ TU/mL. As in the previous step, 5 µL of the lentiviral vector and 5 µL of E2 were injected into eggs at developmental stage 13. A dose of 5 µL of a lentiviral vector with a titer of 1 × 10^9^ TU/mL is suitable for inhibiting the expression of *Sox3* without affecting the survival of the Chinese soft-shelled turtle. Embryos were collected at stages 13–20 after treatment and stored in paraformaldehyde or liquid nitrogen. The group given E2 and *Sox3*-RNAi co-treatment was named the *Sox3* group, whereas the group only treated with E2 was named the CK group. Each group was collected at different developmental stages and each time point contained 3 repetitions.

### 4.5. Library Construction and Sequencing

The mRNA of all samples was enriched by Oligo (dT) beads and fragmented into short fragments using fragmentation buffer and then reverse transcribed into cDNA by using a NEBNext Ultra RNA Library Prep Kit for Illumina (NEB, Ipswich, MA, USA). The Agilent Bioanalyzer 2100 System was used to assess the library quality (Agilent, Santa Clara, CA, USA). The purified double-stranded cDNA fragments were end repaired, A base added, and ligated to Illumina sequencing adapters (Illumina, San Diego, CA, USA). The ligation reaction was purified with AMPure XP Beads (1.0X). Ligated fragments were subjected to size selection by agarose gel electrophoresis and polymerase chain reaction (PCR) amplified. The resulting cDNA library was sequenced using Illumina Novaseq6000 by Gene Denovo Biotechnology Co. (Guangzhou, China).

### 4.6. Bioinformatics Analysis

Reads obtained from the sequencing machines include raw reads containing adapters or low quality bases, which will affect the following assembly and analysis. Thus, to get high quality clean reads, reads were further filtered by fastp. The clean reads were aligned to the *P. sinensis* reference genome (https://www.ncbi.nlm.nih.gov/genome/?term=Pelodiscus sinensis, PRJNA221645, Pelsin_1.0, accessed on 25 May 2023) by using the software Tophat2 v2.1.1. The DEGs between the *Sox3* group and CK group were identified by R package DEseq2, with false discovery rate (FDR) < 0.05 and log_2_FC (fold change (condition 2/condition 1) > 1 or log_2_FC < −1. The upregulated DEGs showed FDR < 0.05 and log_2_FC > 1, and the downregulated DEGs, FDR < 0.05 and log_2_FC < −1. KEGG pathway mapping was performed using Kobas.

### 4.7. Quantitative Reverse Transcription PCR (qRT-PCR)

Embryonic DNA was extracted, and the sex was identified using PCR-based sex-specific markers. The agarose gel only exhibited amplification products in the female individuals, and no bands were amplified from the male individuals [1]. Then, total RNA was extracted using TRIzol reagent and 1 µg RNA was reverse-transcribed into cDNA using the HiScript^®^ III 1st Strand cDNA Synthesis Kit (+gDNA wiper) (Vazyme). According to the instructions, ChamQTM Universal SYBR^®^ qPCR Master Mix (Vazyme) was used for quantitative reverse transcription PCR (qRT-PCR). In the reaction, Gapdh was used as an internal reference gene, and the relative expression was calculated using the 2^−ΔΔCt^ method.

## 5. Conclusions

In summary, this study proved for the first time that *Sox3* may be required for the sex reversal process in the reptile *P. sinensis*. During the embryonic development of *P. sinensis*, *Sox3* was found to be highly expressed at the initiation of ovarian differentiation. There were 271 continuously downregulated genes and 606 continuously upregulated genes differentially expressed between the CK group and *Sox3* group. The KEGG enrichment analysis of the different sex developmental stages after estradiol and *Sox3*-RNAi treatment showed that the deletion of *Sox3* may induce the activation of some signaling pathways such as the mRNA surveillance pathway, TNF signaling pathway, and Ras signaling pathway, and inhibit many signaling pathways, such as the Wnt signaling pathway, TGF-β signaling pathway, and NOD-like receptor signaling pathway, therefore blocking the normal sexual reversal process E2 induces in *P. sinensis*. These results indicated that *Sox3* played a catalytic role in the process of sex reversal and provides theoretical support for all-male breeding technology of *P. sinensis*.

## Figures and Tables

**Figure 1 ijms-25-00248-f001:**
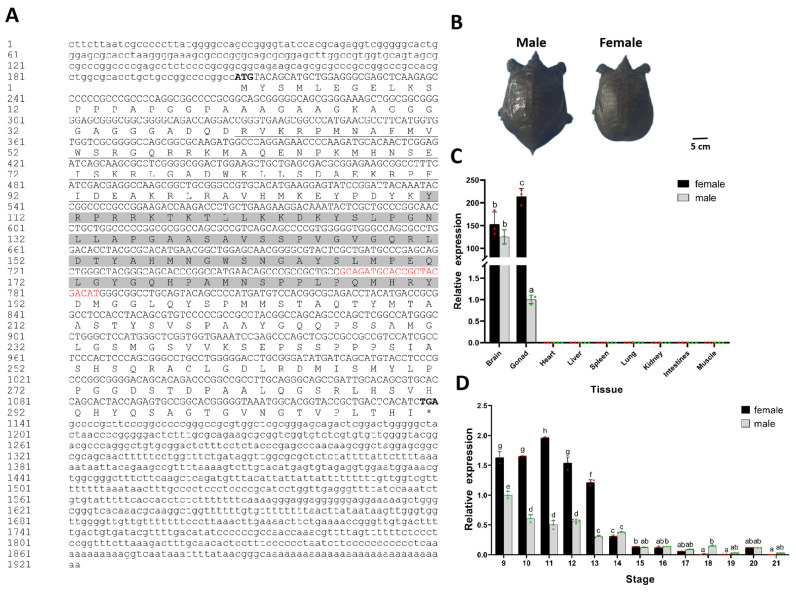
Characterization of Chinese soft-shelled turtle (*P. sinensis*) *Sox3* gene (**A**) Amino acid sequence of *Sox3*. Bold letters denote the starting codon and terminating codon, underscore denotes the HMG-type DNA-binding domain, highlighted sections denote the SOXp motif, red letters denote the conserved fragment of the Sox family. (**B**) Appearance of male and female *P. sinensis*. (**C**) Relative expression levels of *Sox3* in different tissues normed to glyceraldehyde-3-phosphate dehydrogenase (*Gapdh*). (**D**) Relative expression levels of *Sox3* at different developmental stages normed to *Gapdh*. Each value is presented as the mean ± standard deviation of three replicates. One-way analysis of variance with Tukey post hoc tests were used to analyze the means. Different letters indicated a significant difference between male and female embryonic gonads at different tissues or developmental stages (*p* < 0.05).

**Figure 2 ijms-25-00248-f002:**
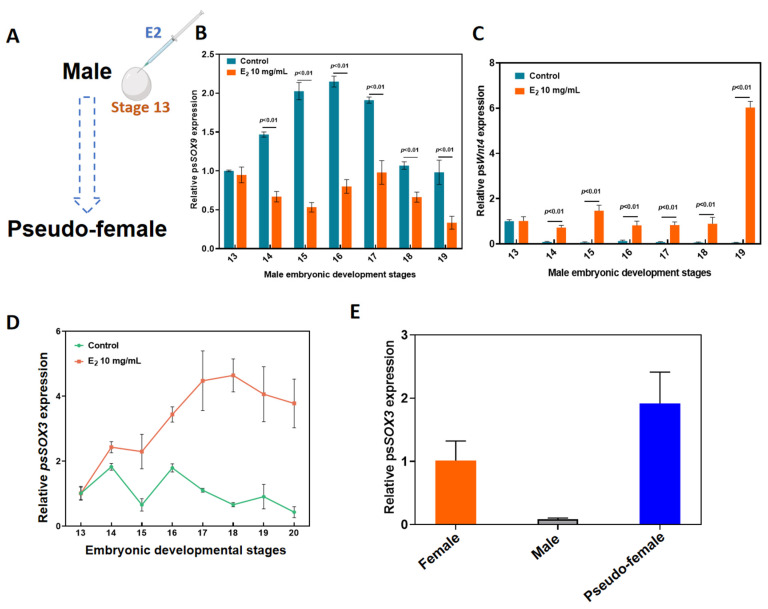
Molecular response of *Sox3* to E2-induced male-to-female sex reversal in *P. sinensis*. (**A**) Method of injection of exogenous hormone E2. (**B**) Expression levels of Sox9 after E2 induction at different developmental stages of male embryos. (**C**) Expression levels of Wnt4 after E2 induction at different developmental stages of male embryos. (**D**) Expression levels of *Sox3* after E2 induction at different developmental stages of male embryos. (**E**) Expression levels of *Sox3* in different types of adult *P. sinensis*. Each value is presented as the mean ± standard deviation of three replicates.

**Figure 3 ijms-25-00248-f003:**
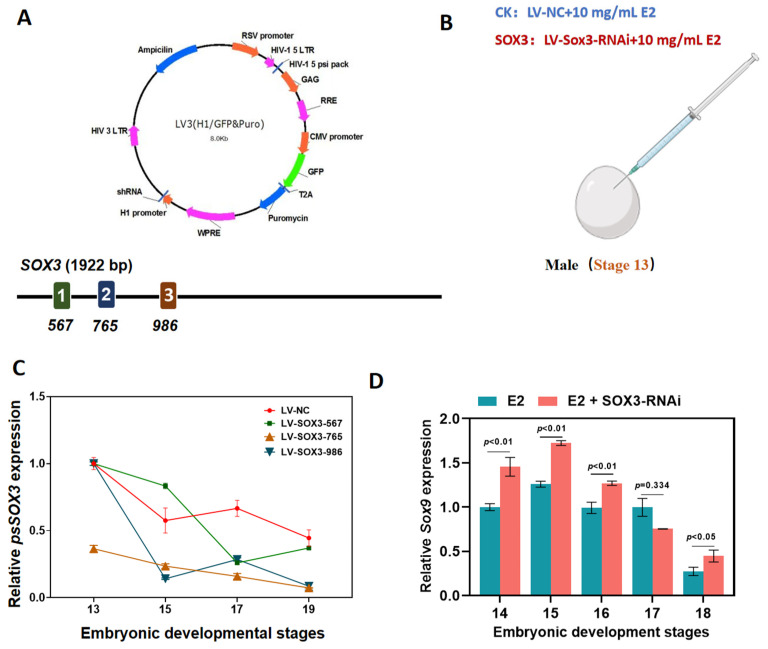
Molecular response of male *P. sinensis* embryos following *Sox3* knockdown. (**A**) Lentiviral interference vector and target locations. Different colors (1, 2, 3) indicate different target locations. (**B**) Method of injection of exogenous hormone E2 and lentiviral interference vectors. (**C**) Expression levels of *Sox3* in embryos injected with E2 and lentiviral interference vectors targeting different target sequences and control embryos. (**D**) Expression levels of *Sox9* in embryos injected with E2 and the lentiviral interference vector Lv-*Sox3*-765 and control embryos. Each value is presented as the mean ± standard deviation of three replicates. One-way analysis of variance with Tukey post hoc tests were used to analyze the means.

**Figure 4 ijms-25-00248-f004:**
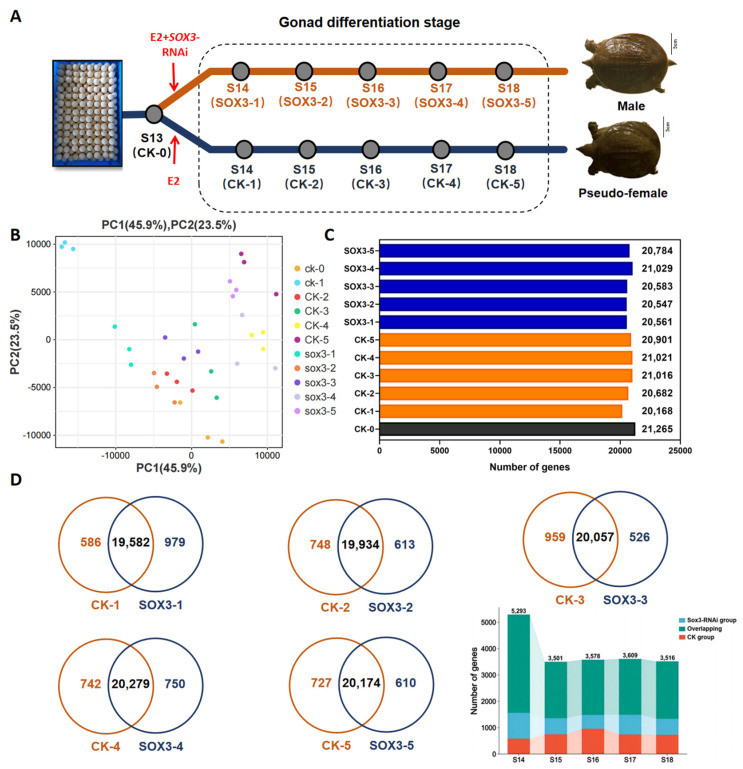
Transcriptome analysis of identified genes at five gonadal differentiation stages during the sex reversal process of *P. sinensis* after *Sox3* knockdown using RNAi. (**A**) Sampling points for embryos induced with E2 and *Sox3*-RNAi (*Sox3*-1 to *Sox3*-5) and embryos induced with only E2 (CK-0 to CK-5). E2 exposure induces the transformation of males into pseudo-females, whereas the male individuals treated by E2 and *Sox3*-RNAi continue to develop into a male. (**B**) Principal component analysis of all data. (**C**) Number of identified genes in the samples. (**D**) Wayne analysis of genes at different time points. Blue represents the identified genes of *Sox3* group (CK-1 to CK-5). Orange represents the identified genes of CK group (CK-1 to CK-5), and black represents the identified genes of CK-0.

**Figure 5 ijms-25-00248-f005:**
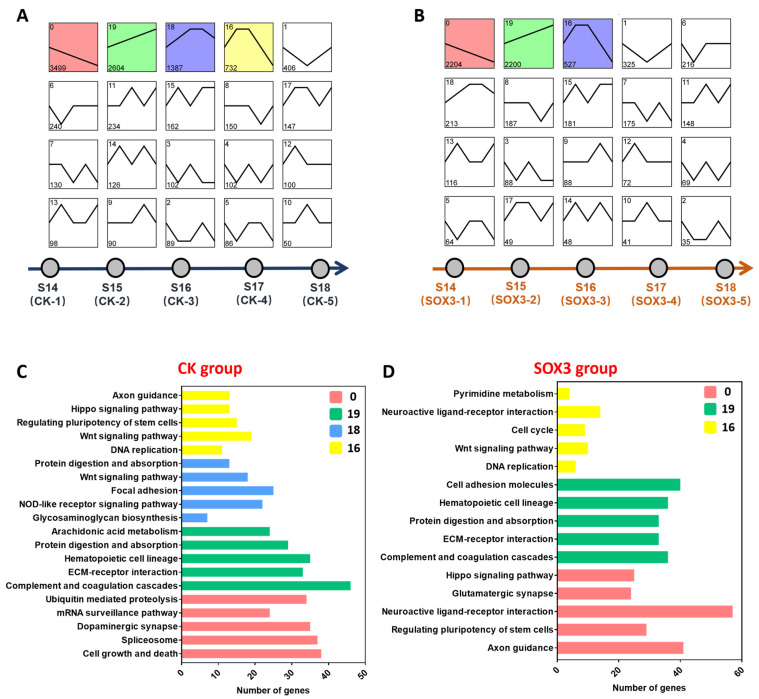
Trend analysis and KEGG enrichment analysis of identified genes after *Sox3*-RNAi treatment. (**A**) Trend analysis of expressed genes identified in the control (CK) groups that were treated with E2. (**B**) Trend analysis of expressed genes identified in the *Sox3* groups. (**C**) Top 5 signaling pathways of important classes of the CK group. (**D**) Top 5 signaling pathways of important classes of the *Sox3* group.

**Figure 6 ijms-25-00248-f006:**
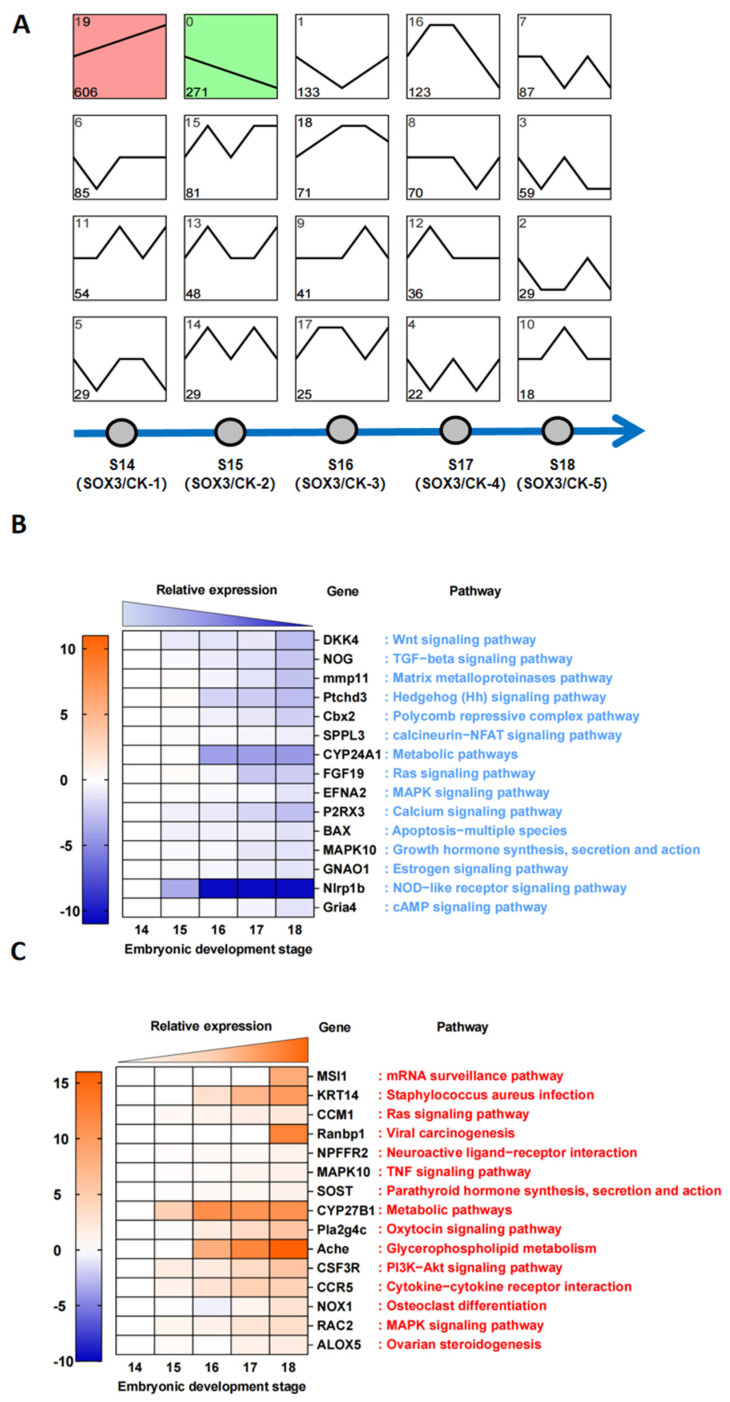
Trend analysis and KEGG enrichment analysis of differentially expressed genes between the control (CK) groups and *Sox3* groups in *P. sinensis*. (**A**) Trend analysis of differentially expressed genes between the CK groups and *Sox3* groups. (**B**) Analysis of expression trends and signaling pathways of top 15 downregulated genes between the CK and *Sox3* groups. (**C**) Analysis of expression trends and signaling pathways of the top 15 upregulated genes between the CK and *Sox3* groups.

**Figure 7 ijms-25-00248-f007:**
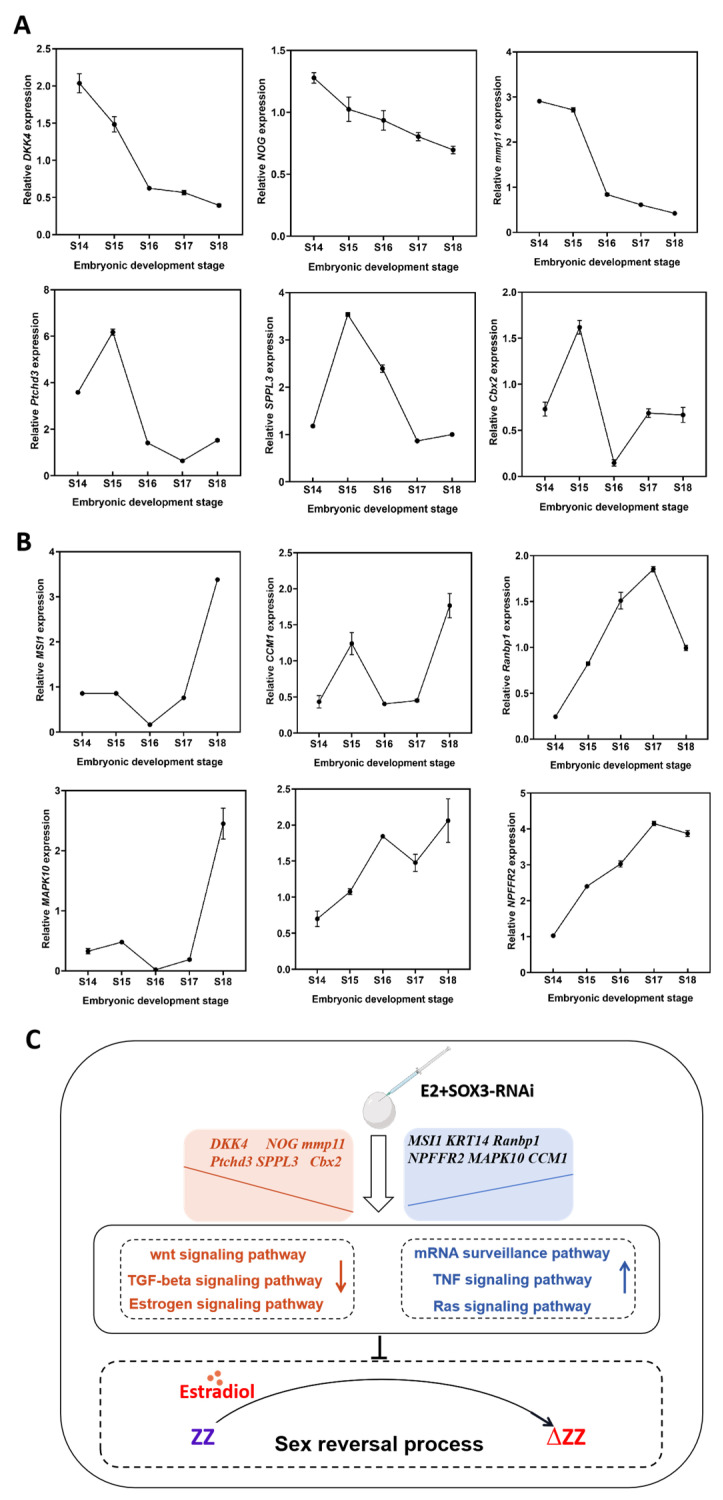
Relative expression levels of key differentially expressed genes in *P. sinensis*. (**A**) Relative expression levels of downregulated differentially expressed genes. (**B**) Relative expression levels of upregulated differentially expressed genes. (**C**) The role of *Sox3* in the sex reversal process. Each value is presented as the mean ± standard deviation of three replicates. One-way analysis of variance with Tukey post hoc tests were used to analyze the means.

## Data Availability

The datasets presented in this study can be found in online repositories. The names of the repository/repositories and accession number(s) can be found below: Genome Sequence Archive, accession number: CRA013318.

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
