# Peer review of "Role of Sox3 in Estradiol-Induced Sex Reversal in Pelodiscus sinensis"

_ijms, 2023, doi:10.3390/ijms25010248_

Round 1
Reviewer 1 Report
Comments and Suggestions for Authors
General comments
This study has reported for the first time that Sox3 may be required for the sex reversal process in the reptile P. sinensis. The study is well-written and materials, methods, and results sections are well described. The authors have conducted many experimental approaches to confirm their findings. However, the authors need to respond to the following points before the approval of publication.
Specific comments
Language revision is required
Line 25, correct “maybe lead” to “may lead”
Line 28, correct “maybe play” to “may play”
Line 425, correct :may be induce” to “may induce”
In abstract
Novelty of this study should be highlighted in abstract
In introduction
Lines 33-35, please indicate the ecological and economic importance of Pelodiscus sinensis
In results
Lines 102-113, “P. sinensis” and all scientific or latin names should be italicized and also should be confirmed throughout the text.
In materials and methods
Specification of injected hormonal dose and timing of examination are very critical approach to obtain an optimal effect. The author should describe the criteria or referring to the reference for selection dose and timing aspects for all hormonal treatment cases (such as Exogenous E2, and Sox3 Lentiviral Interference Vector)
In discussion,
- The author should indicate the points of application of this study, either the application of such knowledge in other reptiles or other animals and for which purposes?
- The discussion is mainly based on repeating the methods and results and no deep discussion points for the obtained results.
Comments on the Quality of English LanguageModerate editing of English language required
Reviewer 2 Report
Comments and Suggestions for Authors
The article sought to elucidate the potential role of Sox3 in E2-induced sex reversal in P. sinensis. In general, the research is well-conducted. However, I have some comments below:
1. Line 120: “specificity” should be replaced with ” biased”
2. Figure 1: Explain the red letters and the highlighted sections in the figure
3. Figure 3: C and B, should have the same font-sizes at the X and Y axes.
4. Figure 4A: By different colors, do the authors mean E2 alone leads to male genotypes and E2+Sox3i leads to pseudo-female? This should be clarified in the figure legends. The genotypes should be indicated during treatment and at the end of the gonad differentiation stage.
5. The use of CK as a control to represent the E2-treated group can create readability challenges. To enhance clarity, I suggest incorporating "E2" where applicable for smoother reading. For instance, consider using "E2+Sox3i
6. Figure 4D:. Are some of the overlapping genes significantly expressed? If so, filtering those genes and finding how many overlapped in all cases (development stages) will be interesting. The overlapping genes should be explained in the legend. Also, it will be interesting to know the total number of genes that are commonly differentially expressed in CK and SOX3 groups (E.g., in SOX3-1 to 5; how many genes are shared between 950, 567, 473, 487 and 546). This will reduce the number of genes, but the key gene will be retained. A list of these genes can be added to the supplementary data.
7. Line 250: Was the gene expression analysis done in this section also carried out using the normal development stages (with samples in Figure 1)? This comparison might help us understand expression patterns better.
8. Line 375: Justify the use of use of stage 13.
9. The authors should check the entire document and ensure scientific names and genes are italicized where necessary.
10. Significant differences indicated in figures must be consistent.
11. The research design and the number of groups, and replicates should be explicitly stated in the materials and methods. E.g., CK was not defined in the materials and methods.
12. How sexes are determined should be clearly stated in the M&M.
Comments on the Quality of English LanguageMinor editing is required
Reviewer 3 Report
Comments and Suggestions for Authors
Author Response
Please see the attachment.

Round 2
Reviewer 1 Report
Comments and Suggestions for Authors
The authors responded and answered all my comments correctly and did great effort to improve the quality of manuscript. Indeed, these revisions have improved the quality of this manuscript significantly. However, I still ask authors to do overall language revisions and editions.
- Page 12, line 307 “maybe play” should be corrected to “may be played” or “may play”
- Sox3, Sox9, Sry, ……. should be italicized only in case of referring to gene and not italicized when meaning the protein.
- All abbreviations should be written in full names when firstly mentioned.
Comments on the Quality of English LanguageMinor editing of English language required
Reviewer 2 Report
Comments and Suggestions for Authors
The authors have enhanced the manuscript. I have a few comments that need to be addressed below:
Line 215 : “E2 can induce the male turns into the pseudo-female” Consider replacing with “E2 exposure induces the transformation of males into pseudo-females,”
Lines 247, 254: “treated by” should be “treated with”
Line 405: “The stage 13 is the key period before the gonadal differentiation stage incubated…” Consider rephrasing the sentence. I suggest “Stage 13 represents a crucial phase in the gonad development stage that precedes gonad differentiation when incubated at a constant temperature.
Line 451: “…only exhibited application products…” Do the authours mean “…only exhibited amplification products…”
Line 450-1: should read “… identified using a PCR-based sex-specific makers,
Comments on the Quality of English LanguageA final check is needed.
